# Impact of Leavening Agent and Wheat Variety on Bread Organoleptic and Nutritional Quality

**DOI:** 10.3390/microorganisms10071416

**Published:** 2022-07-14

**Authors:** Lauriane Mietton, Marie-Françoise Samson, Thérèse Marlin, Teddy Godet, Valérie Nolleau, Stéphane Guezenec, Diego Segond, Thibault Nidelet, Dominique Desclaux, Delphine Sicard

**Affiliations:** 1SPO, University Montpellier, INRAE, Institut Agro, 34060 Montpellier, France; lauriane.mietton@inrae.fr (L.M.); therese.marlin@inrae.fr (T.M.); teddy.godet@inrae.fr (T.G.); valerie.nolleau@inrae.fr (V.N.); stephane.guezenec@inrae.fr (S.G.); diego.segond@inrae.fr (D.S.); thibault.nidelet@inrae.fr (T.N.); 2IATE, University Montpellier, INRAE, Institut Agro, 34060 Montpellier, France; marie-francoise.samson@inrae.fr; 3DIASCOPE, University Montpellier, INRAE, Institut Agro, 34130 Mauguio, France; dominique.desclaux@inrae.fr

**Keywords:** sourdough, yeast, lactic acid bacteria, fermentation, gluten, bread aromatic profile

## Abstract

Leavened bread can be made with different wheat varieties and leavening agents. Several studies have now demonstrated that each of these factors can play a role in bread quality. However, their relative impact in artisanal bread making remains to be elucidated. Here, we assessed the impact of two wheat varieties as well as the impact of sourdoughs and yeasts on multiple components of bread organoleptic and nutritional quality. Using a participatory research approach including scientists and bakers, we compared breads leavened with three different sourdoughs and three different commercial yeasts as well as a mix of sourdough and yeast. Breads were made from two wheat varieties commonly used in organic farming: the variety “Renan” and the landrace “Barbu”. Except for bread minerals contents that mostly depended on wheat variety, bread quality was mostly driven by the fermenting agent. Sourdough breads had lower sugar and organic acids contents. These differences were mostly attributable to lower amounts of maltose and malate. They also had a higher proportion of soluble proteins than yeast breads, with specific aroma profiles. Finally, their aroma profiles were specific and more diverse compared to yeast breads. Interestingly, we also found significant nutritional and organoleptic quality differences between sourdough breads. These results highlight the value of sourdough bread and the role of sourdough microbial diversity in bread nutritional and organoleptic quality.

## 1. Introduction

Bread is an ancient fermented product that is still a staple food in many countries. It can be made with a wide variety of grains and leavening agents. A widely spread and common way to make bread is to use *Saccharomyces cerevisiae* commercial yeast starters and wheat flour. However, recently, global changes and the increasing incidence of non-communicable diseases related to modern diet (including type 2 diabetes, obesity, and food allergies) have led to a renewed interest in sourdough breads and their local production. The appeal of traditionally prepared breads is supported by research showing the nutritional benefits of sourdough bread under laboratory conditions. However, the relative impact of the wheat variety and the leavening agent on the quality of artisanal bread remains to be studied. There is also a lack of knowledge about the differences in quality between traditional sourdough breads.

Traditional sourdoughs, also called type I sourdoughs, are obtained by mixing flour and water, within which microbial communities composed of yeasts and lactic acid bacteria species (LAB) develop. Microbial species diversity in sourdoughs has been well described across the world in bakery-made [1,2,3] and home-made [4] sourdoughs. Sourdough microbial communities are composed of one or two dominating yeast species and up to three lactic acid bacteria. Most sourdough yeast species are members of the family *Saccharomycetaceae* and belong to the genera *Saccharomyces* and *Kazachstania*. Frequently encountered lactic acid bacteria (LAB) include species in the *Lactobacillus, Fructobacillus, Weissella, Leuconostoc,* and *Pediococcus* genera [3]. The proportion of yeast species to LAB species is generally 1:100 with a mean population density of 1.10^7^/1.10^9^ CFU per g of sourdough [1,3]. Acidity in a mature sourdough, ready to be used as a leavening agent, ranges between pH 3.7 and 4.2 [3]. Sourdough strains of *S. cerevisiae* have been shown to be generally different from commercial strains of *S. cerevisiae* both genetically and phenotypically when analysed in laboratory conditions [5]. However, to our knowledge, no studies have compared the effect of sourdough and commercial yeast fermentation and wheat varieties on both organoleptic and nutritional quality of bread in traditional bakery conditions [6,7,8,9,10,11]. Da Ros et al. compared a laboratory-made sourdough bread with a yeast bread and showed that sourdough enhanced the synthesis of short-chain fatty acids and gamma amino butyric acid. Moreover, sourdough increased the value of in vitro protein digestibility, the bread protein quality, and the content of resistant starch and decreased the predicted glycemic index. They also showed that feeding volunteers with sourdough and yeast bread only slightly affects the colon microbiota. Rizzello et al. (2019) compared the digestibility of bread made with a laboratory-made sourdough, a commercial yeast, and a mix of the two and revealed differences in in vitro nutritional indices and in vivo post-prandial markers of gastrointestinal function. In another study, Shewry et al. compared bread made with wheat, emmer and spelt flour, a commercial starter sourdough, and a commercial yeast. They showed that sourdough increases organic acids and mannitol. They also showed that fructan and raffinose contents were decreased by fermentation, with yeast having the greater effect. This effect changed depending on the type of flour, and the type of cereal impacted the total fiber content. Katsi et al. compared the organoleptic properties of breads made with a single sourdough and bread made with a single yeast using three different extraction rates for wheat flour. They showed that sourdough bread had less alcohol and more aldehyde and organic acid than yeast bread, along with a longer shelf life. Finally, Xu et al. 2019 compared bread made with a combination of lab and yeast strains to bread made with yeast strains and showed that sourdough bread had a more complex profile, especially due to having more esters.

During dough fermentation, flour enzymes together with leavening agent microorganisms transform flour components into metabolites that may impact bread quality [12,13,14]. Many cereals can be used to make flour, among which wheat is one of the most commonly used. Wheat flour is composed of starch (ca. 70–75%), water (ca. 14%), proteins (ca. 10–12%), non-starch polysaccharides (ca. 2–3%), lipids (ca. 2%), and soluble carbohydrates such as maltose, sucrose, and glucose (1.55–1.85%) [15]. Flour starch is degraded into glucose and maltose by flour ɑ-amylase, β-amylase, and glucoamylase and by the ɑ-amylase and glucoamylase of some sourdough LABs [16]. Maltose is degraded into glucose by LAB maltose phosphorylase [17] and by maltase (alpha-D-glucosidase) in Saccharomyces yeasts. In dough, the most readily available carbon source is thus maltose, followed by sucrose, glucose, and fructose, along with some trisaccharides such as maltotriose and raffinose. Yeasts transform hexoses into ethanol and CO_2_, while sourdough LABs transform hexoses into lactic acid. In many cases, sourdough LABs are heterofermentative (F. sanfransiscensis, L. brevis, L. rossiae, …) or facultatively heterofermentative (*L. paralimentaius, L. plantarum, L. curvatus*, …) and produce lactic acid, but they also produce acetic acid, ethanol, and CO_2_ from hexoses and pentoses such as arabinoxylans. Homofermentative sourdough LABs (e.g., *L. delbrueckii*) only produce lactic acid from hexoses. Moreover, LABs also produced exopolysaccharides that enhanced bread polysaccharides contents [12].

After starch, proteins are the second major flour component (8–18% dry matter in wheat [18]). In dough, flour proteins form with water a gluten network, which is an association of gliadins (25–80 kDa) and glutenin polymers (>100 kDa) linked by covalent intra- and intermolecular disulphide bonds as well as hydrogen bonds and also formed through hydrophobic interactions [18]. α-, β-, γ-gliadins, classified according to their decreasing mobility in acidic-polyacrylamide gel electrophoresis [19], are low-molecular-weight proteins that are similar in structure and amino acids composition. ω- gliadins differ by their amino acid compositions and their higher molecular weight (60–80 kDA). In wheat grain, glutenins are present in the form of polymers composed of high-molecular-weight (between 67 and 88 kDa) glutenins subunits (HMW-GS) and low-molecular-weight (36 and 44 kDa) glutenins subunits (LMW-GS), representing 60 to 80% of glutenins. During fermentation, LAB and yeast proteases cut gluten disulphide bonds in the presence of a reducing agent, with maximum activity at pH = 3.8 [18].

In dough, aromatic precursors, mostly amino acids, are generated coincidentally by flour enzymes and microorganisms [20]. Bread volatile compounds include higher alcohols, aldehydes, terpenes, organic acids, and esters [21,22,23,24]. In addition to the baking process, which mainly influences the typical aroma of the bread crust, the most important step in the development of crumb flavour is dough fermentation and, among the different variations of the process [20], the type of leavening agent used may have a significant impact [25]. As a matter of fact, enzymatic and microbial conversion of flour components determines bread organoleptic quality [26] and varies according to the different metabolisms of yeasts and LAB strains. Heterofermentative LABs influenced the flavour of sourdough breads, which can be measured with the fermentation quotient Q (i.e., the molar ratio between lactic acid and acetic acid) [27]. Interaction between LAB and yeast may also affect aroma production metabolism [28,29].

In dough, minerals initially present in flour, such as potassium (K), magnesium (Mg), iron (Fe), and zinc (Zn), are mainly chelated with phytate, which reduces their bioavailability [30]. Sourdough is more efficient than yeast fermentation alone in reducing bread phytate content [31]. In addition to LAB and yeast phytase activity, acidification by LAB helps to stimulate flour endogenous phytase. To a lesser extent, some LAB strains, including *L. brevis* and *L. plantarum* strains, also produce phytase [32].

Therefore, the transformation of flour metabolites and thus bread quality may change according to the wheat variety and to the fermenting agent [33]. In this paper, we compared the sugars, organic acids, proteins, minerals, and aroma contents of breads made with eight leavening agents (three sourdoughs, three commercial yeasts, and two mix). We studied the relative impact of wheat variety and leavening agent on each component of bread quality and compared the effect of different leavening agents.

## 2. Materials and methods

### 2.1. Sourdough Origin and Microbial Characterization

Traditional sourdoughs, also called type I sourdoughs, were collected from three bakers in the Occitanie region (France) in January 2020 (Bakers “CRA” from the Cravirola collective, “STE” from Stephane Marrou, “EDI” from Edith Brissiaud). All sourdough were paste-like sourdoughs with a dough yield (DY) between 150 and 250 [34] (EDI—DY = 168; CRA—DY = n.a.; STE—DY = 200). Upon reception in the lab, yeast and bacteria enumerations were performed on YPD and MRS-5 agar media, respectively, as described in Lhomme et al. 2016 [35]. Briefly, 10 g of each sourdough was diluted in TS (0.1% tryptone, 0.85% NaCl) and mixed for 2 min in Stomacher (AES Laboratory, Tours, France). Cascade dilutions were performed from 10–3 to 10–7 and plated in duplicates with a spiral plater (Interscience, Saint-Nom-la-Bretèche, France) on Yeast Peptone Dextrose (YPD) agar (4 g/L Yeast Extract, 8 g/L glucose, 6.8 g/L agar) and MRS 5 containing 0.1 g/L of cycloheximide [36]. A total of 24 LAB and 24 yeast clones were randomly selected from appropriate dilution. Yeast and LAB species were, respectively, identified after ITS rDNA and 16S rDNA PCR amplification and Sanger sequencing, as described in Michel et al. 2016 [37] and Urien et al. 2019 [38].

### 2.2. Flour Preparation

Two semi-wholemeal wheat flours (T80) were obtained with an Astrié stone mill (Ferme de Thuronis, Alaigne, France, 10 January 2020) from either the common variety “Renan” or the landrace “Barbu” and were used to make bread (Appendix A). Ash contents were, respectively, 1.07% dm and 1.05% dm (Appendix A). Renan was chosen among the varieties regularly recommended in the annual list published by the ANMF (Association Nationale de la Meunerie Française) and used by the industrial bakery industry. The Renan wheat grains were a mix of three harvests (2016, 2017, 2019), organically grown in Mauguio (France). Barbu refers here to a unique wheat that can hold several names, such as Barbu du Roussillon and Barbu du Pyrée, due to its geographical origin. This ancient landrace Barbu was chosen from other wheat populations frequently used by farmer-bakers and bakers in the Occitanie region (southern France). The Barbu wheat grains were obtained from moulin Pomairol (Talairan, France) and Minoterie du Pays de Sault (Espezel, France) and were harvested in 2017 after being organically grown in Mauguio (France). The characterization of protein fractions (from F1 to F5, according to Morel et al. [39]) of the two flours was indicated in Appendix A.

### 2.3. Bread-Making Process

Bread-making process was conducted in the bakery “Le pain Levain” (Azillanet, France) on 21 January 2020, using the two flours (Renan, Barbu) and eight different leavening agents (Appendix A). The three type I sourdoughs described above, three commercial yeasts (Hirondelle fresh yeast—Lesaffre “HIR’’, Bioreal organic fresh yeast—Europ-Labo “Bioreal”, and Saf-instant yeast—Lesaffre “Inst”), two mixed leavening agents combining both Hirondelle fresh yeast, (“HIR’’) and sourdough (“STE” or “CRA’’) were used.

Doughs were made by kneading each flour (either Renan or Barbu) with 14.6 g of salt and 0.80 kg of water per kg of flour before the addition of the leavening agent. The six sourdough doughs were made by adding either 180 g of CRA sourdough, 180 g of EDI sourdough, or 180 g of STE sourdough per kg of dough made with either Renan or Barbu flour. The six yeast doughs were made by adding either 4 g of “HIR”, 6 g of “Bioreal”, or 1.4 g of “Inst” yeast per kg of dough made with either Renan or Barbu flour. The four mixed doughs (CRA*HIR and STE*HIR) were made by mixing 1 kg of dough with sourdough (respectively, CRA or STE), 1 kg of plain dough (only salt, mater, and flour) and 5 g of “HIR” yeast for each of the two types of flour (40.5 g of sourdough and 2.7 g of “HIR” per kg of final dough) (Figure 1, Appendix A).

Doughs were left for a first fermentation for a time ranging from 1 h 20 to 1 h 55. Then, they were divided in 500 g samples, shaped, and left for a second fermentation that lasted 3h35. Dough loaves were then baked in a wood-fired oven at 250 °C for 22 to 29 min. For each flour type and each leavening agent, three breads were baked, although only two breads were baked for the two sourdough*HIR yeast mixes (“STE*HIR”, “CRA*HIR”). A total of forty-four breads were thus obtained, sliced, and frozen to be kept at −20 °C.

The breads made with sourdough were named “sourdough breads” and include “CRA breads”, “EDI breads”, and “STE breads”. The breads made only with commercial yeasts were termed “yeasts breads” and corresponded to “Bioreal breads”, “HIR breads”, and “Inst breads”. “CRA*HIR breads” and “STE*HIR breads” were the breads made with both sourdough and yeast.

### 2.4. Bread Proteins Quantifications

Bread protein quantifications were performed on each of the 44 breads (2 wheat varieties*8 leavening agent *[2 breads for CRA*HIR and STE*HIR or 3 breads for CRE, EDI, STE, Bioreal, HIR, Inst]) in triplicates. Two to three slices of approximately 1 cm thickness were cut in the middle of each bread. The crust was removed, and crumb pieces were individually packed in sealed plastic bags and stored at −20 °C until freeze-drying. Slices were then ground with an A10 basic mill (IKA, Staufen, Germany). The moisture content of the resulting powder was determined according to AACC method 44-15.02, and the total protein content was determined using the Kjeldhal method according to AACC 46-12.01, with 5.7 used as conversion factor.

Proteins were extracted from bread, according to Morel and Bar-L’Helgouac’h [39]. A total of 160 g of bread sample was suspended in a 20 mL sodium phosphate buffer (0.1 M pH 6.9) containing 1% (*w*/*v*) SDS (sodium dodecyl sulfate) and stirred for 80 min at 60 °C. After centrifugation (39,000× *g*; 30 min; 20 °C) the supernatant containing soluble proteins was stored (−20 °C) until SE-HPLC analysis. The pellet was resuspended in SDS-sodium phosphate buffer containing 20 mM dithioerythritol (DTE) before being sonicated for 3 min at 3.5 watts. The new supernatant was stored until analysis. A residual pellet still remained containing all the unextracted proteins.

Proteins recovered after the different extraction steps were separated by size-exclusion high-performance liquid chromatography (SE-HPLC) using a TSKgel G4000 SWXL column (7.8 mm i.d. × 30 cm, TOSO BIOSCIENCE GmbH, Griesheim, Germany), according to Dachkevitch and Autran (1989), on an Alliance system (Waters, Milford, MA, USA). Proteins were eluted at ambient temperature with 0.1 M sodium phosphate buffer (pH 6.9) containing 0.1% (*w*/*v*) SDS at a flow rate of 0.7 mL/min, and absorbance was recorded at 214 nm. The first chromatogram, corresponding to SDS-soluble proteins, was divided into five fractions, from F1 to F5, according to Morel et al. [39]. Areas under fractions F1 to F5 were summed up to represent SDS-soluble proteins. The total area under the second chromatogram represents the protein fraction extracted after DTE reduction and sonication (DTE-extracted proteins, also called insoluble proteins, or FI fraction). The total protein content of samples was estimated as indicated by Morel et al. [39]. The non-extracted protein fraction was calculated by subtracting the sum of SDS-soluble protein content and DTE-extracted protein content from the Kjeldahl total protein content. When the non-extracted protein fraction value was negative, due to high recovery, this value was forced to 0, and the sum of SDS-soluble and DTE-extracted protein fractions corrected to reach 100%.

### 2.5. In Vitro Bread Digestibility Test

In vitro bread digestibility was measured for each of the 44 breads (2 wheat varieties*8 leavening agents *[2 breads for CRA*HIR and STE*HIR or 3 breads for CRE, EDI, STE, Bioreal, HIR, Inst]) in triplicates. Digestibility of the bread samples was evaluated by measuring the rate of in vitro proteolysis using the Protein Digestibility Assay kit (Neogen, Auchincruive, UK) with the following slight modifications: for each sample, proteolysis was performed in replicates on 250 mg of bread. At the end of the 3 h trypsin/chymotrypsin digestion, the digestion was stopped by immersing the tubes in boiling water, and the tubes were cooled and then centrifuged at 4696× *g* for 15 min at 15 °C. After centrifugation, the supernatants were set aside and the pellets frozen. The rate of proteolysis was then estimated by determining the amount of nitrogen remaining in the pellets from two extractions of the same sample by the Kjeldahl method. The proteolysis rate after 1 h of peptic digestion, followed by 3 h of tryptic/chymotryptic digestion, is the % of remaining protein calculated from the total protein content of the starting sample.

### 2.6. Bread Sugars, Organic Acids, and Alcohols Contents Analysed with HPLC

Bread sugars, organic acids, and alcohols contents were measured in each of the 44 breads (2 wheat varieties*8 leavening agent *[2 breads for CRA*HIR and STE*HIR or 3 breads for CRE, EDI, STE, Bioreal, HIR, Inst]). Bread sugars, organic acids, and alcohols contents were analysed using HPLC. For each bread slice, 5 g were sampled and homogenised in 10 mL of distilled water in a stomacher (AES laboratoire, France). Supernatants were centrifuged at 4500 rpm for 10 min, and 1 mL was clarified with Carrez reagents I and II (125 μL each) and centrifuged at 13,000 rpm for 5 min. HPLC analyses were performed with a HP 1100 LC system (Agilent technologies, Santa Clara, CA, USA) equipped with a refractive index detector (RID Agilent G1382A) and an Agilent G1314A UV detector. Bread sugar, organic acid, and alcohol compounds were measured on a Rezex ROA-organic acids column (SDVB–H+ 8%, 300 × 7.8 mm, Phenomenex, Torrance, CA, USA) under the following conditions: mobile phase 0.005N H2SO4, flow rate 0.6 mL/min, column temperature 60 °C, injection volume 15 µL, run time 25 min. The extract was diluted six-fold with mobile phase before injection in order to obtain an acidic sample. The organic acids contents were obtained thanks to UV detection (wavelength of 210 nm), while the sugars and the alcohols contents were estimated with a RID detector (temperature 35 °C). All concentrations were calculated by external calibration. All products used as standards were HPLC or analytical grade, purchased from SIGMA. Each of them was injected at 3 or 4 levels chosen in accordance with the expected concentration, in the same conditions as samples (preparation procedure, elution program, injection volume).

Simple sugars and polyols refer here to arabinose, fructose, glucose, maltose, mannose, raffinose, mannitol, glycerol, and meso-erythritol. Organic acids refer to acetate, lactate, malate, pyruvate, and succinate, and alcohol refers to ethanol. Xylose was not found in any of the samples and thus will not appear in the results.

### 2.7. Minerals

Minerals contents were measured for each of the 44 breads (2 wheat varieties*8 leavening agent *[2 breads for CRA*HIR and STE*HIR or 3 breads for CRE, EDI, STE, Bioreal, HIR, Inst]) in triplicate. Frozen bread samples were cut, lyophilized, and powdered. As described in Chay and Mari [40], after an acidic digestion (nitric acid and H_2_O_2_), the samples were then placed in a HotBlock for acidic hydrolysis at 85 °C for 24 h. Iron (Fe), zinc (Zn), magnesium (Mg), copper (Cu), calcium (Ca), potassium (K), phosphorus (P), and sodium (Na) were separated and quantified with a microwave Plasma Atomic Emission Spectroscopy (Microondes Speedwave, PS-AES Agilent). Results are expressed in µg/g of the element per gram of dry bread.

### 2.8. Volatile Compounds Analysis

Volatile compounds analysis was performed on each of the 44 breads (2 wheat varieties*8 leavening agent *[2 breads for CRA*HIR and STE*HIR or 3 breads for CRE, EDI, STE, Bioreal, HIR, Inst]). A semi-quantitative analysis of bread aroma compounds was carried out by HS-SPME-GC-MS. For each slice of bread, three 1.5 g crumb samples were placed into SPME vials and spiked with 0.75 μg 4-nonanol internal standard (Merck, Darmstadt, Germany). Samples were incubated and agitated for five min at 60 °C on a CombiPal autosampler (CTC Analytics AG, Zwingen, Switzerland), and volatile compounds were extracted in the headspace with a SPME triphased Divinylbenzene Carboxene Polydimethylsiloxane fiber (Supelco, Bellefonte, PA, USA) for 1 h at 60 °C. Volatile compounds adsorbed on the fiber were desorbed in the injector of a gas chromatograph (GC Trace, Thermoscientific, Waltham, MA, USA) for 3 min at 250 °C in splitless mode. Extracted volatile compounds were analysed with a gas chromatograph coupled with a mass spectrometer. Compounds separation was performed on a DBWAX column (Agilent Technologies, Santa Clara, CA, USA), dimensions 30 m, 0.25 mm, 0.25 μm, with helium BIP (Air Products, Allentown, PA, USA) as carrier gas and at a constant flow rate of 1.2 mL/min. The following oven parameters were used for this analysis: initial temperature was 40 °C held for 3 min, followed by an increase at a rate of 6 °C/min to 245 °C, this last temperature being held for 5 min. Triple quadrupole mass spectrometer TSQ8000 (ThermoScientific, Waltham, MA, USA) was used as a simple quadrupole and operated in scan mode over a mass range from 46 to 206 amu in electronic impact mode at an ionization potential energy of 70 eV. Ion source and transfer line were, respectively, set at 230°C and 250 °C. Compounds were identified by comparison of the spectra with NIST (Gaithersburg, MD, USA) and Wiley (John Wiley & Sons, Inc., Hoboken, NJ, USA) databases, and only compounds presenting a match factor > 800 were retained. The internal standard was used to semi-quantify volatile compounds based on peak area ratio.

Thirty-eight aroma compounds were analysed, of which ten alcohols (1-hexanol, 1-octen-3-ol, 1-pentanol, 2-phenylethanol, 2-methyl-1-butanol, 2-methyl-1-propanol, 3-methyl-1-butanol, 3-nonen-1-ol-(E), benzyl alcohol, ethanol), 9 aldehydes (2-heptenal, 2-hexenal, 2-octenal-(E), 3-methyl-butanal, benzaldehyde, heptanal, hexanal, nonanal, octanal), 3 acids (acetic acid, hexanoic acid, octanoic acid), 4 esters (ethyl_heptanoate, ethyl hexanoate, ethyl octanoate, hexyl acetate), 4 ketones (1-octen-3-one, 2-octanone, 4-nonanone, 6-methyl-5-hepten-2-one), 3 furan derivatives (2-pentylfuran, 5-methyl-2-furfural, furfural), 2 lactones (2-nonenoic acid gamma-lactone, gamma-nonalactone), 2 phenol derivatives (vanilline, 4-vinylguaiacol), and 1 terpene (limonene), specifically, were found in yeast and sourdough breads, according to the literature [23,29,41,42,43].

### 2.9. Statistics

A total of twenty-eight quantitative variables were analysed to characterize bread nutritional quality: total bread protein content (%dm), SDS-soluble protein proportion (% of total protein content), DTE-extracted protein proportion (% of total protein content), non-digested protein proportion (% of total protein content), bread sugars and polyols contents in g/kg of bread (arabinose, fructose, glucose, maltose, mannose, raffinose, mannitol, meso-erythritol, glycerol), bread organic acids contents in g/kg (acetate, glutarate, lactate, malate, pyruvate, succinate), bread alcohols content in g/kg (ethanol), and bread minerals content in ppm (Fe, Zn, Mg, Cu, Ca, K, P and Na). In addition, thirty-eight aroma compounds were quantified.

All analyses were carried out on three bread replicates (three different breads made from the same dough). For each bread, three samples (three different pieces) of the same bread were analysed for minerals and protein contents, while only one sample was studied for sugars, organic acids, alcohols, and aroma compounds.

The factors accounting for the variation in each variable among leavening agents and wheat varieties were analyzed with the following mixed analysis of variance model:Y_ijkl_ = *μ* + *α*_i_ + *β*_j_ + *γ*_ij_ + D_k_ + *ε*_ijkl_,(1)
where Y_ijkl_ is the quantitative variable, *α*_i_ is the fixed leavening agent effect, *β*_j_ is the fixed wheat variety effect, *γ*_ij_ is their interaction effect, D_k_ is the bread random effect, and *ε*_ijkl_ is the residual error. For the quantitative variables for which we had no technical replicates within bread, the bread random effect was not included in the model. When included, the bread random effect D_k_ was never significant and will therefore not be presented. Significance of differences between means was assessed using Tukey’s HSD method. *p*-values were all corrected for multiple testing using the Bonferroni correction.

All statistical tests were performed using R software, version 4.0.3.

## 3. Results

In this work we compared nutritional and organoleptic quality of breads made from two wheat varieties and eight different leavening agents (three type I sourdoughs, three yeast starters, and two mixes). Overall, a total of 44 breads were made. Yeast breads and sourdough breads were made in triplicates, while sourdough*yeast mixed breads were made in duplicates. For each bread, carbohydrate content, protein content, and aromatic profiles were analyzed.

### 3.1. Leavening Agent Characteristics

The three Type I sourdoughs used to make the experimental breads were collected from different bakeries known to have divergent sourdough microbial communities. This was confirmed by the analysis of the final sourdoughs used to make bread with the culture-dependent method (Figure 2, Appendix A). Sourdough “CRA” contained *Kazachstania bulderi* and *Fructilactobacillus sanfranciscensis* as dominant yeast and LAB species, respectively. Sourdough “STE” contained *Saccharomyces cerevisiae* and *Torulaspora delbrueckii* as co-dominant yeasts and *F. sanfranciscensis* as dominant LAB. Sourdough “EDI” contained both *Kazachstania unispora* and *S. cerevisiae* as main yeast species, with *Companilactobacillus paralimentarius, Lactobacillus spicheri, Lactiplantibacillus plantarum*, and *F. sansfranciscensis* as co-occurring LAB species (Figure 2). LAB population densities were, respectively, 4.06 × 0^8^, 1.6 × 10^9^, and 7.8 × 10^9^ colony-forming units (CFU) per g of sourdough for CRA, EDI, and STE, respectively, while yeast population densities were 2.46 × 10^7^, 4.38 × 10^7^, and 5.00 × 10^3^ per gr of sourdough (Appendix A). For the three commercial yeasts: Bioreal, HIR, and Inst yeast population densities were, respectively, 5.9 × 10^9^, 4.0 × 10^9^, and 2.2 × 10^10^ CFU per g (Appendix A).

A total of 900 gr sourdough was added to 5 kg of dough to make sourdough bread, leading to total numbers of LAB cells per gr of dough of 6.2 × 10^7^, 2.4 × 10^8^, and 1.2 × 10^9^ and total numbers of yeast cells per gr of dough of 3.7 × 10^6^, 6.7 × 10^6^, and 7.6 × 10^2^. Respectively, 6 g, 4 g, and 1.4 g of Bioreal, HIR, and Inst were added per kg of yeast bread doughs, leading to total numbers of yeast cells per g of dough of 2.9 × 10^5^, 2.4 × 10^5^, and 3.8 × 10^5^.

### 3.2. Bread Carbohydrate, Polyol, and Organic Acid Contents

We first compared the total carbohydrates, polyol, and organic acid content of bread (Appendix A). Yeast breads had higher total sugar, polyol, and organic acid contents than sourdough and mixed breads (Figure 3, Appendix A). Significant differences between sourdough breads were also detected. Breads made with EDI sourdough had a higher total sugar and polyol content than the other sourdough breads. Similar results were found for total organic acid content, except that no differences were detected among sourdough breads (Figure 3, Appendix A). Bread total alcohol content (ethanol and glycerol) was not significantly different between yeast and sourdough breads (Figure 3, Appendix A). In addition to the leavening agent, wheat variety also had a significant impact on the total carbohydrate, polyol, and organic acid content of the bread. Barbu breads had significantly higher contents of total sugar, total alcohol, and total organic acids than Renan breads (Appendix A).

We then analyzed the contribution of each metabolite to bread variation by carrying out a principal component analysis on all sugar, organic acids, and alcohol metabolites contents (Figure 4). The first two axes explained 56.52% of variance, with 33.91% in the first dimension and 22.62% in the second. The first axis separated yeast breads from sourdough and mixed breads, while the second axis separated breads according to their wheat variety. Maltose, fructose, succinate, glycerol, malate, meso-erythritol, acetate, lactate, and mannitol variables mainly contributed to the first axis, while ethanol, glucose, glutarate, and pyruvate mostly contributed to axis 2.

The statistical analysis of each metabolite separately confirmed these observations.

The leavening agent significantly impacted the content of each carbohydrate metabolite except glutarate and arabinose (Appendix A). The three yeast breads had a significantly higher maltose, fructose, and raffinose average content (47.67 +/− 0.18, 7.95 +/− 0.06, 6.25 +/− 0.09, 6.05 +/− 0.10 mg/g, respectively) compared to sourdough and mixed sourdough/yeast breads (Appendix A). They also had a significantly higher average malate content (12.18 +/− 0.06 g/kg). Between sourdough breads, EDI always had a significantly higher amount of maltose, fructose, raffinose, and malate (38.78 +/− 0.36, 4.77 +/− 0.04, 7.94 +/− 0.20, 2.70+/− 0.07 mg/g, respectively). It also had a lower amount of glucose (4.65 +/− 0.11 mg/g). Mixed breads had an intermediate average amount of each carbohydrate except for mannitol, fructose, acetate, and mannose (Figure 3, Appendix A).

Wheat variety also significantly impacted the content of metabolites, except for arabinose, glutarate, glycerol, lactate, maltose, mannose, meso-erythritol, and succinate. With regards to the most abundant metabolites (maltose, raffinose, malate, glucose, and fructose, with content over 5 g/kg of bread), Barbu breads contained significantly more raffinose, glucose, and fructose (7.81 +/− 0.05, 5.26 +/− 0.07, 5.16 +/− 0.14 mg/g, respectively) than Renan breads (Appendix A).

### 3.3. Bread Soluble and Insoluble Protein Content

The analysis of total protein content of the 44 breads revealed that both the leavening agent and the wheat variety significantly contributed to its variation (Appendix A). However, most variation was found between breads made with different wheat varieties. Renan breads had significantly more proteins (9.90% dm +/− 0.06) than Barbu breads (9.31% dm +/− 0.04) (Appendix A).

We then compared the proportions of soluble proteins (SDS_soluble), insoluble proteins (DTE_extract proteins), and non-extracted proteins (Non_extracted) within the total protein content in the different types of breads. Breads made with the eight leavening agents and the two wheat varieties had similar proportions of non-extracted proteins (Non_extracted = 7.94% +/− 0.55). By contrast, they differed in soluble and insoluble proteins proportion (Appendix A). Yeast breads had the lowest average proportion of soluble proteins (SDS_soluble = 17.48% +/− 0.19) and the highest average proportion of insoluble proteins (DTE_extract = 73.75% +/− 0.95, Appendix A, Figure 5), while sourdough breads had the highest average proportion of soluble protein (SDS_extract = 29.37% +/− 1.31, Appendix A) and lowest proportion of insoluble proteins (DTE_extract = 64.11% +/− 1.19, Appendix A). Mixed breads were intermediate (SDS_soluble = 25.21% +/− 0.54, DTE_extract = 65.63% +/− 1.28, Appendix A). Among sourdough breads, a lower proportion of soluble proteins was found for EDI sourdough bread (SDS_soluble = 23.00% +/− 0.94, Appendix A) than for CRA and STE sourdough breads (34.19% +/− 1.64 and 30.94% +/− 0.50, respectively, Appendix A).

Regarding the wheat effect, breads made with Barbu wheat had significantly higher soluble proteins proportion than breads made with Renan (Appendix A). Note, however, that this effect depended on the leavening agent. Barbu breads made with STE*HIR leavening agent contained significantly less soluble proteins than Renan breads (Appendix A).

Finally, we analyzed in-vitro bread digestibility. The remaining protein fraction after proteolysis (Non_digested fraction) ranged from 69.8% to 74.9% with a mean of 73.2% +/− 0.56. (Appendix A). There was no significant leavening agent effect on the remaining protein fraction. However, there was a wheat variety and a wheat variety*leavening agent interaction effect (Appendix A). Breads made with the Renan variety had a better in vitro protein digestibility than Barbu (Appendix A). Surprisingly, no significant correlations between bread digestibility and DTE_insoluble protein fraction (r = −0.03, *p* = 0.8) or between digestibility and SDS_soluble protein fraction (r = 0.15, *p* = 0.34) were found.

### 3.4. Bread Mineral Content

We then analyzed bread mineral content by following eight minerals: iron, zinc, magnesium, copper, calcium, potassium, phosphorus, and sodium (Appendix A). Na, K, and P were the most abundant minerals with, respectively, 5,74 +/− 0.07 mg, 2.89 +/− 0.11 mg, and 2.13 +/− 0.03 mg per g of bread (Appendix A). Most of the observed variation was explained by the wheat variety (Figure 6). Breads made with Barbu had significantly more Fe, Zn, Mg, Cu, Ca, and K than breads made with Renan that had more P and Na (Appendix A).

The effect of the leavening agent on the mineral content was also significant but less clear (Figure 6: Na, K, P). There was no apparent trend according to the type of leavening agent (yeast, sourdough, or mix of the two). For example, STE sourdough breads had a significantly higher Na content than all yeast (Bioreal, HIR, and Inst) and mixed breads (CRA*HIR, STE*HIR) (F_7,116_ = 8.43; *p*-value < 0.0001), while the other sourdough breads did not. Yeast breads and sourdough breads CRA and EDI had a higher P content than the sourdough bread STE and mixed breads (F_7,116_ = 18.77; *p*-value < 0.0001) (Appendix A).

### 3.5. Bread Aroma Profiles

A total of 38 aroma compounds (10 alcohols, 9 aldehydes, 3 acids, 4 esters, 3 furan derivatives, 4 ketones, 2 lactones, 1 terpene, and 2 phenol derivatives) were analyzed in the crumbs of the 44 breads. The most abundant aroma compounds were furfural, which is known to have a “sweet, woody, almond, bread, baked” odor, and 3-methyl-1-butanol, which is an alcohol with a “pungent, ethereal, cognac, banana, molasses” odor [41,42,44,45,46]. Higher amounts of furfural were found in sourdough breads, while higher quantities of 3-methyl-1-butanol were found in yeast breads.

Sourdough breads had specific aroma profiles compared to yeast breads (Appendix A). Aroma profiles also varied according to the sourdough. CRA and STE sourdough breads and mixed breads had a wider diversity of aroma compounds with a higher relative content. EDI’s sourdough breads had an aroma profile closer to yeast breads than to other sourdough breads (Figure 7). Generally, alcohols were found to be more abundant in yeast breads (2-methyl-1-propanol, 2-phenylethanol, 3-methyl-1-butanol, 2-methyl-1-butanol, 4-vinylguaiacol, and 1-octen-3-ol). By contrast, aldehydes, ketones, esters, furan derivatives, acids, vanillin, terpene, and other alcohols (benzyl-alcohol, 3-nonen-1-ol-(E), 1-hexanol, and 1-pentanol) were more abundant in sourdough breads (Figure 7). Interestingly, mixed breads, by sharing characteristics of both sourdough breads and yeast breads, had a wider diversity of aroma compounds.

We found that 12 out of the 38 aromatic compounds were significantly impacted by both the leavening agent and the wheat variety (1-hexanol, 1-octen-3-one, 1-pentanol, 2-heptenal, 3-nonen-1-ol-(E), 6-methyl-5-hepten-2-one, acetic acid, benzyl alcohol, furfural, gamma-nonalactone, hexanoic acid, octanal, octanoic acid), 8 by only the leavening agent (2-hexenal, 2-methyl-1-propanol, 3-methyl-1-butanol, 4-nonanone, 4-vinylguaiacol, 5-methyl-2-furfural, benzaldehyde, hexanal) and only 1 by only the wheat variety (heptanal) (Appendix A).

## 4. Discussion

In this work, we assessed the differences in nutritional and organoleptic quality between breads made with eight different leavening agents and two different wheat flours. Previous studies also compared sourdough and yeast breads [6,7,8,9,10,11]. The originality of this work is that type I sourdoughs were used, two wheat varieties were tested, and real artisanal hand-made bread-making processes were carried out, with a total fermentation time of five hours on average. This study is therefore complementary to others testing sourdough benefits in laboratory conditions.

We investigated both sugar and gluten bread contents. Both FODMAPs (Fermentescible oligosaccharides disaccharides monosaccharides and polyols) and gluten are suspected to be involved in irritable bowel syndrome [46,47,48,49,50,51]. Previous studies have investigated how sourdough may affect FODMAPs content [52,53,54]. Here, we show that sourdough breads had lower maltose, fructose, and polyols contents than yeast breads. We also found an increased mannitol content in sourdough breads and mixed breads. This reveals a nutritional advantage of sourdough bread over yeast bread, as 75% of ingested mannitol is fermented by the intestinal flora, and mannitol is considered a low-caloric sweetener (glycemic index of 0) [55]. The lower level of sugar in sourdough bread may be related to the presence of the maltose-consuming LAB *F. sanfranciscensis* in sourdoughs. In addition, it may be explained by the metabolic activity of *S. cerevisiae*, which can also assimilate maltose through its isomaltose gene (IMA) and maltose gene cluster (MAL). The yeast maltose gene cluster is composed of three genes encoding the maltose transporter (permease; MAL1), maltase (MAL2), and transcriptional regulator (MAL3). A previous study revealed that sourdough strains of *S. cerevisiae* have a higher copy number of MAL and IMA genes and deplete maltose more efficiently [5] than commercial yeast starters. Therefore, the genetic differences between sourdough and commercial starters of *S. cerevisiae* may also explain at least part of the differences in maltose content between sourdough and yeast breads. In the same way, the lower fructose content in sourdough breads can be explained by the presence of *F. sanfranciscensis* that uses fructose as an electron acceptor to produce acetate, while *S. cerevisiae* produces fructose by the action of invertase. Finally, the lower malate content in sourdough breads can be explained by the fact that most LAB are able to convert malate into lactate via the malolactic fermentation reaction: malate→lactate + CO_2_ [43].

Regarding gluten, we show that sourdough bread contained an increased ratio of SDS-soluble protein over DTE-extracted protein in sourdough bread, suggesting another nutritional benefit of using sourdough as leavening agent. Surprisingly, we did not find in vitro protein digestibility differences between sourdough and yeast breads. Previous studies have found that sourdough fermentation increased in vitro protein digestibility compared to yeast fermentation [56]. In vivo analysis of sourdough bread digestibility also revealed a better digestibility of sourdough bread [56]. Our results partly contradict these previous ones. An in vitro protein digestibility assay kit does not account for the complexity of digestible metabolic activity. Additional experiments in the Human Intestinal Microbial Ecosystem simulator (SHIME) and in vivo trials should be performed to compare artisanal sourdough and yeast bread digestibility in conditions closer to human gut. The fact that yeast breads had a lower SDS-soluble proteins proportion and a higher insoluble protein proportion (DTE-extracted and non-extracted proteins) than sourdough breads might be explained by the higher acidity of sourdough breads, which promotes protease activity, and by LAB protease activity per se that is absent in yeast breads. Indeed, LAB proteolytic activity is highly efficient due to cell-wall-associated proteinases and intracellular peptidases [9,54]. This is to be compared to yeast and cereal proteases that only participate in the first stage of proteolysis, which reduces gluten disulfide bonds to form variously sized polypeptides [56]. Moreover, in bakeries, the fermentation time is often longer for sourdough bread than for yeast bread, and so is the protein proteolysis time. This could lead to a higher level of soluble proteins content in sourdough bread compared to yeast breads [9,57].

We also observed that yeast and sourdough breads had specific aroma profiles, which confirmed previous studies [45,58,59]. Sourdough breads had a richer aroma profile than yeast breads, as well as having more aroma compounds found in higher quantities, confirming previous results that showed a greater content of volatiles in sourdough breads compared to breads chemically acidified with lactic and acetic acids [20,59]. Specifically, acids, aldehydes, esters, ketones, and furan derivatives were higher in sourdough breads. Conversely, iso-alcohols (2-methyl-1-propanol, 2-methyl-1-butanol, and 3-methyl-1-butanol) and particularly 3-methyl-1-butanol, considered as a specific bread aroma compound, originating from amino acids degradation by microorganisms [45], were most present in mixed breads and yeast breads.

In addition to the two main bread-making processes that use either Type I sourdough or commercial yeast, there are other processes for making bread rise. Some bakers use a yeast starter together with a type I, II, or III sourdough. This bread can be called “bread with sourdough”. The nutritional and organoleptic properties of sourdough, as well as the ease with which starter can be used to leaven the dough, have recently led to a strong revival of interest in “ bread with sourdough”. However, the nutritional and organoleptic benefits of making bread with sourdough and yeast have not been studied. The interaction between the microbial community of the sourdough and the yeast starter could, however, reduce some of the nutritional values of the sourdough bread, or, conversely, the sourdough could inhibit some of the valuable properties of the yeast starter such as leavening or the production of flavoring compounds. To investigate the interaction between type I sourdough and yeast starter, we analyzed the nutritional and organoleptic quality of what we called “mixed bread”, i.e., bread made with half sourdough and half yeast dough. Except for mannitol, acetate, mannose, and fructose, mixed breads had intermediate metabolite content compared to yeast and sourdough breads showing a simple additional effect of sourdough microbial community and yeast starters for most metabolites. However, mannitol and acetate contents were higher and fructose and mannose content lower in mixed breads than in sourdough and yeast breads. Mannitol is a fructose-derived alcohol produced mostly by heterofermentative LAB via the mannitol-dehydrogenase phosphoketolase pathway [60,61,62]. A previous study [62] showed an increase in mannitol when LAB were co-cultured with yeast. This can be attributed to the maintenance of redox balance but also to the competition with *S. cerevisiae* for glucose as the main carbon source with an increase in citrate co-metabolism [63]. In addition, the presence of *S. cerevisiae* favored acetic acid production [64]. The decrease in fructose in mixed bread also revealed that mixing sourdough and yeast starter could be an interesting alternative to reduce FODMAP. Additional experiments on the interaction between sourdough microbial community and yeast starters are required to obtain further insight on the role of microbial interaction in bread carbohydrate content.

Beyond differences between yeast breads, mixed breads, and sourdough breads, we detected differences in carbohydrate contents and aroma profiles between sourdough breads. CRA and STE sourdoughs contained the heteromentative *F. sanfranciscensis* as dominant LAB species, while EDI sourdough harboured *C. paralimentarius* and *L. plantarum* as dominant species (with *F. sanfranciscensis* only at low frequencies). *F. sanfranciscensis* consumes maltose and glucose [60] and transforms them into CO2, lactate, ethanol, and acetate in the presence of an electron acceptor such as fructose [13]. To achieve redox balance, fructose is converted to mannitol via the Mannitol-dehydrogenase phosphoketolase pathway [60,61], and malate is converted to lactate with malolactic enzyme [13,61]. By contrast, the dominant LAB species in EDI are facultative heteromentative LAB that do not transform fructose into mannitol and malate into lactate, explaining why CRA and STE sourdough breads showed significantly lower maltose and fructose content and higher mannitol and lactate contents. Differences among sourdough breads can also be explained by differences in yeast species and population density. EDI sourdough breads had sugars and polyols content, soluble/insoluble protein proportion, and aroma profile closer to yeast breads than other sourdough breads (2.7 × 10^7^ CFU per g of yeast dough on average and 6.7 × 10^6^ CFU per g of EDI dough). This is consistent with EDI sourdough containing *S. cerevisiae* as the dominant species (also present in STE sourdough but at a much lower yeast density: 5.0 × 10^3^ CFU per g in STE and 4.4 × 10^7^ CFU per g in EDI).

The majority of the differences in nutritional and organoleptic quality of the breads were related to the leavening agents. However, flours contributed to the differences in mineral content of breads. They also contributed to the aromatic differences in breads, as illustrated by the fact that the aroma varies significantly depending on the flour and therefore the variety of wheat used. They also contributed to the differences in raffinose, glucose, and sugar content. Many organic bread-making food chains are exploring the diversity of wheat varieties, with the aim of maintaining cereal diversity in the field and of increasing the potential of adaptation to climate change. This is the case of the wheat landrace “Barbu” that we used here, which is an ancient variety that was abandoned at the end of the 19th century and replanted in the south France “Occitanie” region recently. The effect of pre- and post-industrial wheat varieties on the nutritional and sensory quality of breads has already been studied [65,66,67]. As we have shown here, previous studies have found that wheat genotypes impact the bread mineral content and sensory profile of bread. The results of this paper provide additional information by presenting the relative impact of the leavening agent and the wheat variety on bread quality. We show that the leavening agent is the main driver of bread quality, regarding protein content, sugar content, and aroma profiles, while wheat varieties are important for mineral content and influence but to a lesser extent than the other bread quality parameters.

In conclusion, this study highlighted the benefits of artisanal sourdough bread by showing that yeast breads had higher sugars, organic acids, and alcohol contents and a lower proportion of soluble/insoluble proteins than sourdough breads. Yeast breads did not differ significantly from each other, while sourdough breads, which were made with sourdoughs having different microbial communities, did. This result highlights the relevance of analyzing functional diversity of sourdough yeast and bacteria in future research.

## Figures and Tables

**Figure 1 microorganisms-10-01416-f001:**
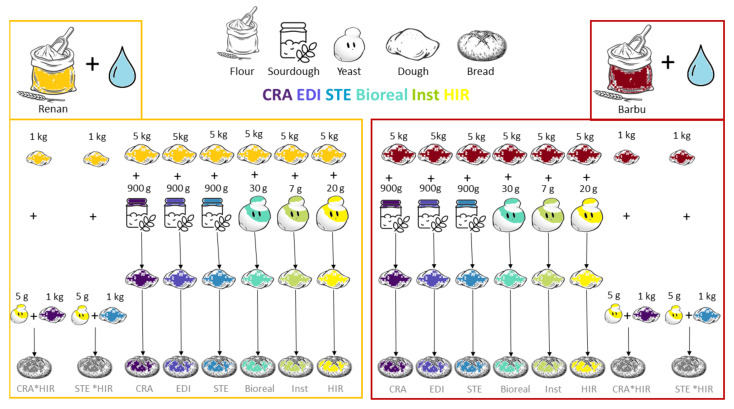
Bread-making design of the experiment. “CRA”, “EDI”, and “STE” refer, respectively, to the bakers Collectif Cravirolla, Edith Brissiaud, and Stéphane, and “HIR”, “Bioreal”, and “Inst” refer, respectively, to the three commercial yeasts Hirondelle fresh yeast—Lesaffre, Bioreal organic fresh yeast—Europ-Labo, and Saf-instant instant yeast—Lesaffre.

**Figure 2 microorganisms-10-01416-f002:**
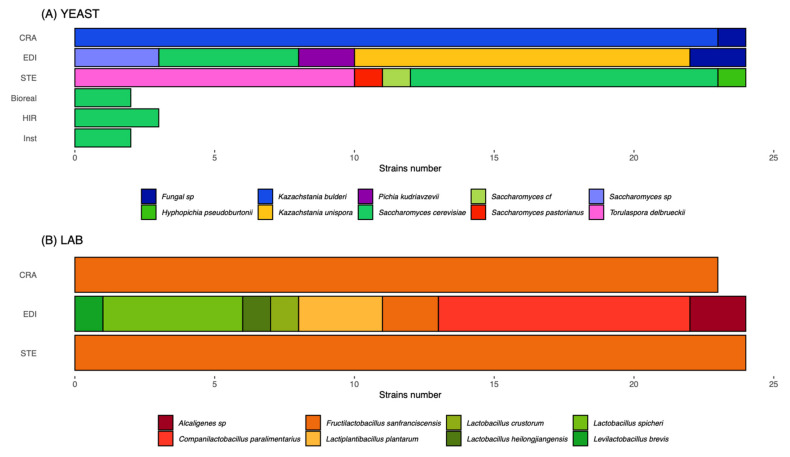
Diversity of yeast and LAB species in the three sourdoughs (“CRA” from Collectif Cravirolla, “EDI” from Edith Brissiaud, “STE” from Stéphane Marrou) and in the three commercial yeasts (Hirondelle fresh yeast—Lesaffre, “HIR”, Bioreal organic fresh yeast—Europ-Labo, “Bioreal” and Saf-instant instant yeast—Lesaffre, “Inst”). Twenty-four yeast strains were characterized for each sourdough and two to three strains for each yeast starter. Twenty-three or twenty-four LAB strains were characterized per sourdough. Each colour refers to a different species. Each line refers to a different leavening agent. The abscise indicates the number of strains of each species.

**Figure 3 microorganisms-10-01416-f003:**
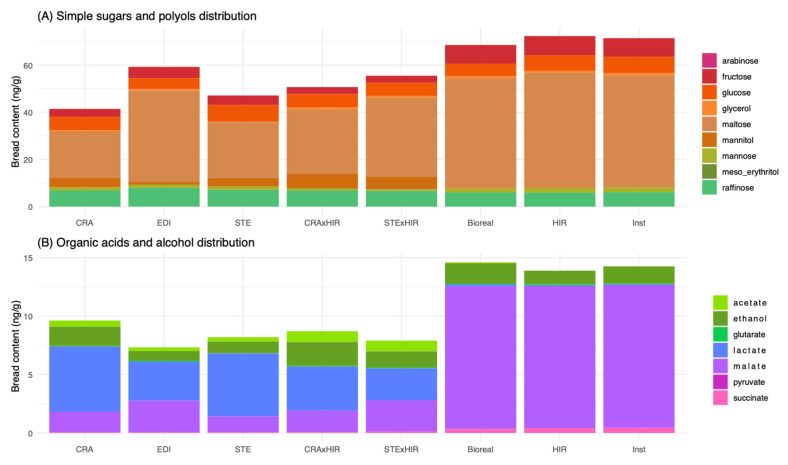
Distributions of (**A**) bread sugars (fructose, glucose, maltose, mannose, raffinose, arabinose, and xylose) and polyols (mannitol, glycerol, and meso-erythritol) contents (g per kg of bread) according to the leavening agent and (**B**) bread organic acids (acetate, glutarate, lactate, malate, pyruvate, and succinate) and ethanol contents (g per kg of bread) according to the 8 leavening agents (sourdoughs CRA, EDI, and STE, mixed sourdough/yeast CRA*HIR and STE*HIR, and commercial yeasts Bioreal, HIR, and Inst). (“CRA”, “EDI”, and “STE” refer, respectively, to the bakers Collectif Cravirolla, Edith Brissiaud, and Stéphane, and “HIR”, “Bioreal”, and “Inst” refer, respectively, to the three commercial yeasts Hirondelle fresh yeast—Lesaffre, Bioreal organic fresh yeast—Europ-Labo, and Saf-instant instant yeast—Lesaffre,).

**Figure 4 microorganisms-10-01416-f004:**
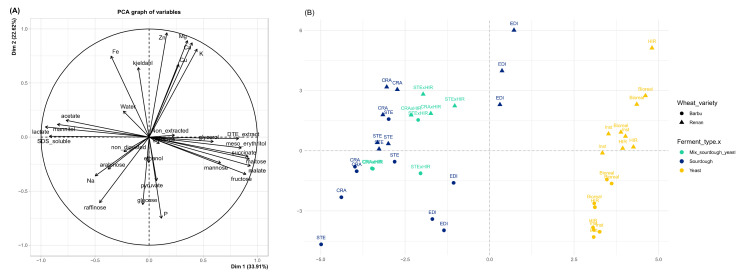
Principal component analysis with (**A**) graph of all quantitative variables according to leavening agent—bread sugars, polyols, organics acids and alcohols contents, bread protein fractions proportions, bread protein total content, non-digested bread protein proportion, and bread minerals contents—and (**B**) plot according to the 8 leavening agents (sourdoughs CRA, EDI, and STE, mixed sourdough/yeast CRA*HIR and STE*HIR, and commercial yeasts Bioreal, HIR, and Inst) and the 2 wheat varieties (Barbu, Renan). (“CRA”, “EDI”, and “STE” referr, respectively, to the bakers Collectif Cravirolla, Edith Brissiaud, and Stéphane, and “HIR”, “Bioreal”, and “Inst” refer, respectively, to the three commercial yeasts Hirondelle fresh yeast—Lesaffre, Bioreal organic fresh yeast—Europ-Labo, and Saf-instant instant yeast—Lesaffre).

**Figure 5 microorganisms-10-01416-f005:**
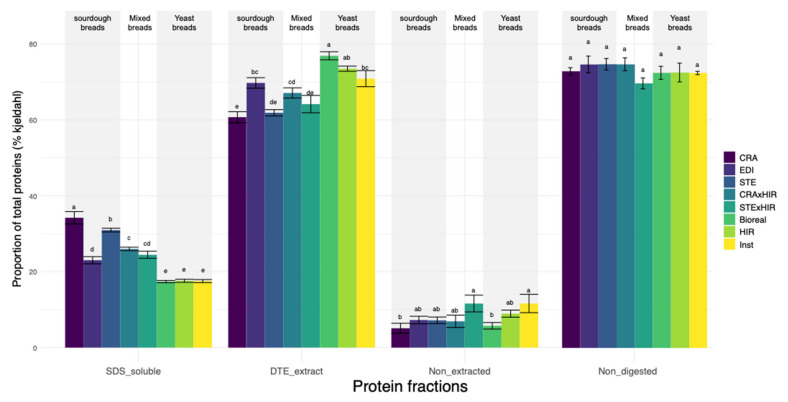
Proportions (out of total protein content—Kjeldahl) of protein fractions (SDS_soluble proteins, DTE-extracted and non-extracted proteins) and in vitro digestibility (Non_digested) of the three sourdough breads (CRA, EDI, STE), the two mixed breads (CRA*HIR, STE*HIR), and the three yeast breads (Bioreal, HIR, Inst). (“CRA”, “EDI”, and “STE” refer, respectively, to the bakers Collectif Cravirolla, Edith Brissiaud, and Stéphane Marrou, and “HIR”, “Bioreal”, and “Inst” refer, respectively, to the three commercial yeasts Hirondelle fresh yeast—Lesaffre, Bioreal organic fresh yeast—Europ-Labo, and Saf-instant instant yeast—Lesaffre). Different letters on top of bars indicate significantly different means.

**Figure 6 microorganisms-10-01416-f006:**
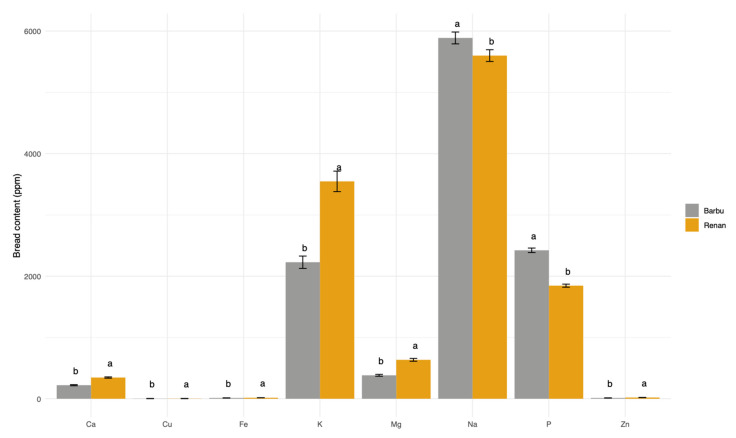
Bread minerals contents in ppm (iron (Fe), zinc (Zn), magnesium (Mg), copper (Cu), calcium (Ca), potassium (K), phosphorus (P), and sodium (Na)) according to the wheat variety (Renan, Barbu). Different letters on top of bars indicate significantly different means.

**Figure 7 microorganisms-10-01416-f007:**
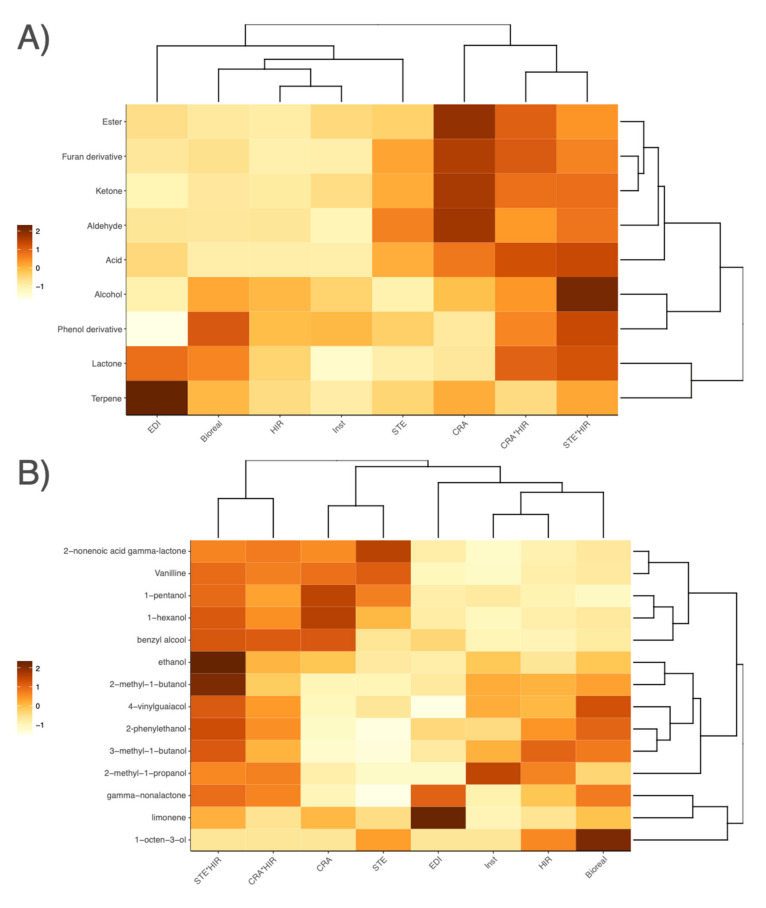
Heat maps in the three sourdough breads (CRA, EDI, STE), the two mixed breads (CRA*HIR, STE*HIR), and the three yeast breads (Bioreal, HIR, Inst) of (**A**) the relative content of the chemical classes of the 38 aroma compounds analyzed and (**B**) the relative content of the aroma compounds whose classes were shared among all the breads (alcohols, lactones, phenol derivative, and terpene). (“CRA”, “EDI”, and “STE” refer, respectively, to the bakers Collectif Cravirolla, Edith Brissiaud, and Stéphane, and “HIR”, “Bioreal”, and “Inst” refer, respectively, to the three commercial yeasts Hirondelle fresh yeast—Lesaffre, Bioreal organic fresh yeast—Europ-Labo, and Saf-instant instant yeast—Lesaffre).

## Data Availability

All data and analysis script are available in open access under Zenodo (https://zenodo.org) with the DOI: 10.5281/zenodo.5876156.

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
