# Peer review of "Impact of Leavening Agent and Wheat Variety on Bread Organoleptic and Nutritional Quality"

_microorganisms, 2022, doi:10.3390/microorganisms10071416_

Round 1

Reviewer 1 Report

Dear authors,

the topic of the submitted manuscript sounds interesting and appropriate methods were used. The presented data revealed difference of product quality due to different leavening agents. The conducted work seems to have a high working load, which should be suitable for publication in a high quality journal. Nevertheless, the submitted manuscript reveled some drawbacks and inconsistencies, which have to be corrected before publication.

The volume of baked breads should be added as main quality criteria used for research. Semi-wholemeal wheat flours are not very common, the ash and protein content should be measured and added. Kernels from 2016 and/or 2017 seem to be “old”, how were the grains stored? The description of recipes and bread making process is hard to understand, this part should be rephrased. Furthermore, a table in supplement with all recipes should be added. In “2.9 Statistics” the number of analyses and replicates are mentioned. This kind of information should be added to each method with the total number of measurements. In general, it is not clear which samples are included in results (each of the 8 different leavening agents made from 2 flours, totally 16 breads)? Even in the tables of supplement only results from leavening agents are listed. Are these values averages from both flours?

Results about in vitro digestability are missing. Moreover, it is implausible that no difference of in vitro digestibility was found between sourdough and yeast breads. Is there any explanation for this result? Fermentation time? The part of microbial analysis is inconsistent, the description is missing in “2. Materials and methods”. Wheat does strain number mean in Fig.1? In Tab. S1 “Density of LAB and yeast in the 8 leavening agents”, but data for commercial yeasts are missing. If it was not analyzed for commercial yeasts, this information should be added (n.a. not analyzed).

Due to missing information, it is not possible to follow the results and discussion. Summing up, the current quality of manuscript is insufficient for publication and has to be improved strongly.

Some minor errors and further recommendations:

  • Fig. 2: resolution should be improved, seems to be a little bit blurry
  • Fig. 4: for easier understanding colours of bars should be selected according to groups of bread (yeast, mixed & sourdough)
  • Fig. 5 (Fig. S2): it’s not clear if mineral composition is from breads (8 different leavening agents and two flours are in sum 16 different breads) or only from the two different flours
  • Fig. S1: average content for all breads made with two different flours? Should be separated in each recipe (each of the 8 different leavening agents made from 2 flours, totally 16 breads)
  • The number of citations seems to be too high and should be checked/reduced

Reviewer 2 Report

The paper “Impact of leavening agent and wheat variety on bread organoleptic and nutritional quality” analyses the impact of wheat variety and leavening agent on the sugars, organic acids, proteins, minerals and aroma contents of bread samples. Two wheat varieties and 8 leavening agents were used for this purpose.

The article topic is of relevance for readers due to both selected influencing factors on bread quality. Also, the quality of artisanal bread is nowadays an important issue for bread makers and consumers.

A large number of analyses were performed and the results were interpreted by relevant statistical methods.

Some observations are:

  1. “microbial communities are composed of. one or two dominating yeast species…” ; please remove the dot after of
  2. I suggest introducing the references from lines 59-60 using an explanatory text
  3. when the fermentation activity of yeast and LAB is discussed in the Introduction section, I suggest completing the information with the formation of exopolysaccharides and the importance of fermentation quotient (lactate/acetate) for the flavor
  4. line 52, complete “CFU” per g of sourdough
  5. line 126, remove the reference from the bracket
  6. Subsection ” Bread-making process”- it is well known that the bread formulation is also a factor affecting its quality in terms of sensorial and nutritional parameters. Please explain why the authors used 80% water addition or 1,46% salt in the formulation?

Also, even if is of crucial importance, the dough yield in the case of sourdough was not mentioned by the authors.

I propose to add an experimental scheme to this section, to better understand the research layout. It will be useful for readers.

  1. Results section

Lines 365-366- “The leavening agent significantly impacted the content of each carbohydrate metabolite save glutarate and arabinose”

Lines 385-386 “Wheat variety also significantly impacted the content of a majority of metabolites save for arabinose, glutarate, glycerol, lactate, maltose, mannose, meso-erythritol and succinate contents”

Are these sentences correct?

  1. Line 548-complete  (3.0 *105 CFU ….
  2. I did not find the results for In vitro bread digestibility test

Reviewer 3 Report

It is well written and worth to be published manuscript in general. 
However, I have a few comments.

Literature data are not cited with accordance of Molecules journal and the manuscript is not formatted according to Instructions for Authors. 
What was the yield of the flour? Flour yield has a significant influence on flour chemical composition and consequently on bread properties. 
Authors precisely characterized many properties of obtained bread but forgot about the characterization of properties of used raw materials. If authors compare the influence of varying the characteristic used two flours should be also included.  
What was the reason to incorporate 0.8 kg of water per kilogram of flour? Did the authors study flour water absorption?
The quality of Fig. 3 is poor and difficult to read. Moreover, all abbreviations should be explained under the figure and tables. 
References are not properly formatted with the guidance of the Authors. 

Round 2

Reviewer 1 Report

Dear Authors,

the manuscript was improved, but only to a limited degree. How results are presented is confusing, a clear structure and order should be adressed. As mentioned (comment 6) protein digestability was included at the end of section 3.3. Nevertheless, results are given only in the supplement, which is inadequate. These results could be added to Fig.5. Furthermore, correlation analysis of protein fractions and digestability should be added (perhabs even for more results). The missing volume of breads is a big drawback of the study and why it was not measured is still questionable (in comment 1 it was only mentioned that volume was not measured). The discuss the quality of bread as affected by fermentation and cultivar is a very important part, which can be done only with strong limitations when the volume of breads is not known.

Fruthermore, some minor corrections should be performed. All figures have poor quality, they should be replaced by ones with superior resolution. Be aware that in vitro has to be written in italic letters, check the whole manuscript.

Discussion has still a low quality and should be improved strongly:

- "In bakeries, the fermentation time is often longer for sourdough bread than for yeast bread and so is protein proteolysis time. This could lead to even more difference in protein content between sourdough and yeast breads." correct and specify "protein content"

- "Some consumers state they will digest sourdough bread better than yeast bread (bakers’ statements)." such general statements have to be avoided in scientific publications! Why you did not cite some studies, which revealed better digestability of sourdough bread, e.g. measured by in vitro methods.

- "The DTE-extracted protein fraction mostly contained proteins with disulfide bonds (i.e. glutenins and alpha, beta and gamma gliadins) involved in gluten-related disorders." most proteins in wheat contain disulfide bonds, even non-gluten proteins; this sentence makes no sense and has to be revised.

- "Here we did not find any difference of in vitro digestibility between sourdough and yeast breads but additional experiments should be performed to compare sourdough and yeast bread digestibility in more realistic conditions." it has to be stated that the obatined results regarding digestability are contradictory to other studies

Author Response

Dear reviewer,

Please find below our responses to all you valuable comments.

the manuscript was improved, but only to a limited degree. How results are presented is confusing, a clear structure and order should be adressed.

Response: We have made many changes to make the structure of the results and discussion clearer and more fluid. We have expanded the discussion and better highlighted the interest of the results of the paper. We have highlighted the main changes in red. We believe this new version is much better and we thank the reviewer for his comment that allowed us to improve our paper significantly.

As mentioned (comment 6) protein digestability was included at the end of section 3.3. Nevertheless, results are given only in the supplement, which is inadequate. These results could be added to Fig.5. 

Response: We have now added the digestibility results to Figure 5

Furthermore, correlation analysis of protein fractions and digestability should be added (perhabs even for more results).

Response: We did not find any significant correlation. We have added this result in the text now. 

The missing volume of breads is a big drawback of the study and why it was not measured is still questionable (in comment 1 it was only mentioned that volume was not measured). The discuss the quality of bread as affected by fermentation and cultivar is a very important part, which can be done only with strong limitations when the volume of breads is not known.

Response: We completely agree that measuring the volume of bread would have been very complementary to our data. Unfortunately, we did not measure it and have no remaining bread.

Fruthermore, some minor corrections should be performed. All figures have poor quality, they should be replaced by ones with superior resolution.

Response: The quality of all figures has been improved

Be aware that in vitro has to be written in italic letters, check the whole manuscript.

Response: We have corrected it

Discussion has still a low quality and should be improved strongly:

Response: We thank the reviewer for this comment. We have improved the discussion.

- "In bakeries, the fermentation time is often longer for sourdough bread than for yeast bread and so is protein proteolysis time. This could lead to even more difference in protein content between sourdough and yeast breads." correct and specify "protein content"

Response: Thank you for this comment. We have replaced the sentence by “This could lead to a higher level of soluble proteins content in sourdough bread compared to yeast bread”.

- "Some consumers state they will digest sourdough bread better than yeast bread (bakers’ statements)." such general statements have to be avoided in scientific publications! Why you did not cite some studies, which revealed better digestability of sourdough bread, e.g. measured by in vitro methods.

Response: Thank you for this comment. Citing consumers is often use in sociology but we agree that it is not usual in biological science paper.  We have now deleted the sentence that cites “consumers”.

- "The DTE-extracted protein fraction mostly contained proteins with disulfide bonds (i.e. glutenins and alpha, beta and gamma gliadins) involved in gluten-related disorders." most proteins in wheat contain disulfide bonds, even non-gluten proteins; this sentence makes no sense and has to be revised.

Response: We thank the reviewer for this remark. We have deleted the sentence.

- "Here we did not find any difference of in vitro digestibility between sourdough and yeast breads but additional experiments should be performed to compare sourdough and yeast bread digestibility in more realistic conditions." it has to be stated that the obatined results regarding digestability are contradictory to other studies

Response: We thank the reviewer for his comment that allow us to improve the discussion.  It is always difficult to interpret negative results. We have now completed the discussion part on gluten and clearly stated that our results are contradictory to others. “Previous studies have found that sourdough fermentation increased in vitro protein digestibility compared to yeast fermentation (Rizzello et al. 2014). In vivo analysis of sourdough bread digestibility also revealed a better digestibility of sourdough bread (Rizzello et al. 2019). Our results partly contradict these previous ones. In vitro protein digestibility assay kit does not account for the complexity of digestible metabolic activity. Additional experiments in Human Intestinal Microbial Ecosystem simulator (SHIME) and in vivo trials should be performed to compare artisanal sourdough bread and yeast bread digestibility in conditions closer to human gut.”
